

# Effective asexual reproduction of a widespread soft coral: comparative assessment of four different fragmentation methods

Sohyoung Kim, Christian Wild and Arjen Tilstra

Marine Ecology Department, Faculty of Biology and Chemistry, Universität Bremen, Bremen, Germany

## ABSTRACT

**Background:** Many coral reefs worldwide are experiencing declines in hard corals, resulting in other benthic organisms, *e.g.*, soft corals, becoming more dominant. As such, more studies on the ecophysiology of soft corals are needed. Despite many methods for asexual reproduction of hard corals, effective methods for soft corals, *i.e.*, without a hard skeleton, are scarce. This study, thus, assessed four fragmentation methods, the glue, rubber band, tunnel mesh, and plug mesh method for the pulsating soft coral *Xenia umbellata* that is widely distributed in the tropical Indo-Pacific.

**Methods:** Methods were comparatively assessed by determining the required time and labor for the fragmentation plus the health status of the fragmented corals by measuring their oxygen fluxes and pulsation rates, *i.e.*, a special feature of this soft coral that can be used as a proxy for its health.

**Results:** There were no significant health status differences between methods. This was indicated by similar gross photosynthesis (between 7.4 to 9.7 $\mu$g O$_2$ polyp$^{-1}$ h$^{-1}$) and pulsating rates (between 35 and 44 pulses min$^{-1}$) among methods. In terms of time/labor intensity and success rates, *i.e.*, the percentage of fragments attached to the desired surface, the plug mesh method was the most efficient method with a significantly higher success rate (95 $\pm$ 5%), while the others had a success rate between 5 $\pm$ 5 and 45 $\pm$ 15%. The time needed for fragmentation, though not significant, was also the shortest (78 $\pm$ 11 s fragment$^{-1}$), while other methods required between 84 $\pm$ 14 and 126 $\pm$ 8 s frag$^{-1}$. The plug mesh method may thus be a valuable tool related to the reproduction of soft corals for use in subsequent experimental work.

Corresponding author
Sohyoung Kim,
soh_kim@uni-bremen.de

## INTRODUCTION

Coral reefs are high biodiversity ecosystems that provide services such as coastal protection, food provision, and tourism which maintain the livelihoods of millions of people (*Moberg & Folke, 1999*; *Reaka-Kudla, 1997*; *Woodhead et al., 2019*). However, these complex ecosystems have been declining for decades due to global and local anthropogenic

stressors (*De'Ath et al., 2012*; *Naumann et al., 2015*; *Hoegh-Guldberg, Pendleton & Kaup, 2019*). These stressors can trigger a so called 'phase shift', where benthic dominance shifts from hard corals towards algae-, sponge-, or soft coral dominance (*Done, 1992*; *McClanahan, Cokos & Sala, 2002*; *Maliao, Turingan & Lin, 2008*; *Wood & Dipper, 2008*; *Wild et al., 2011*; *Baum et al., 2016*). Although phase shifts from hard coral to (macro)algae are studied relatively often, research on the shift towards soft coral dominance are more scarce (*Norström et al., 2009*; *Cruz et al., 2016*).

Though soft coral communities are rising in numbers and softs corals becoming the dominant organism on many reefs, there is a lack of studies exploring the ecological impact of these communities. One study by *Wood & Dipper (2008)* indicated that some soft corals, like xeniids, could decrease the overall biodiversity by not only blocking the settlement and recruitment of hard corals, but also by overgrowing and eventually killing the remaining ones. Not only are they considered invasive due to their high fecundity and long reproductive season (*Wood & Dipper, 2008*), xeniid soft corals are also more resilient to global and local stressors (*Wild et al., 2011*; *Vollstedt et al., 2020*). This could intensify the phase shift from hard coral towards soft coral dominance. As the effects of these shifts are yet to be established, there is an urgent need for more relevant studies assessing soft coral ecophysiology to better understand and manage this benthic functional group accordingly.

When conducting research with soft corals, however, securing sufficient numbers of coral samples is one of the first yet highly time-consuming procedures which could span weeks to months depending on the number of samples. As the samples often need to be similar in size and should be easy to relocate, *i.e.*, should be attached to a small rock or an independently moveable substrate such as a coral plug, many *ex-situ* experiments utilize a method called 'fragmentation'. While there are many detailed and straightforward fragmentation protocols for hard corals, *i.e.*, corals that build a hard calcium carbonate (CaCO$_3$) skeleton (*Johnson et al., 2011*; *Forsman et al., 2015*), it is difficult to adopt the same methods for corals that lack a hard skeleton, *i.e.*, soft corals (*Veron, 2011*). Though some soft corals have a support system that would allow the use of the same methods as for hard corals, *e.g.*, tough leathery tissue as associated with coral genera such as *Sinularia* and *Sarcophyton*, or gorgonians which are supported by a gorgonin skeleton, many other soft corals lack any rigid support, such as the widespread xeniid soft corals. These soft corals, despite being a successful colonizer (*Benayahu & Loya, 1985*), may need special methods as current approaches can be time- and labor-intensive when preparing hundreds of similar sized fragments with minimal disturbances. While much information exists in grey literature and aquarist exchange platforms, there are a limited number of scientific papers exploring the topic of fragmentation for these types of soft corals (*Ellis & Sharron, 1999*). Some studies by *Barton, Willis & Hutson (2017)*, *Borneman & Lowrie (2001)*, and *Chaitanawisuti & Kritsanapuntu (2019)* do touch upon soft coral fragmentation methods, but lack a method for the corals without any rigid support system.

Thus, the aim of this study is to (i) assess and compare four fragmentation methods for the pulsating soft coral, the *Xenia umbellata*, *i.e.*, the glue, rubber band, tunnel mesh, and plug mesh method, and (ii) provide a protocol for the best method. Among the four

methods, the glue and rubber band methods have already been widely used in laboratories (*e.g.*, *Barton, Willis & Hutson, 2017*; *Ellis & Sharron, 1999*; *Simancas-Giraldo et al., 2021*; *Vollstedt et al., 2020*). In contrast, the tunnel mesh and plug mesh methods were modified from a recently patented method named the tunnel cube method by *Curry (2020)*. We assessed the methods by ranking them in four evaluations: success rate, time efficiency, oxygen fluxes, and pulsating rate. The first two were used to determine the labor/time efficiency and the latter two were used to assess the health. After assessments, optimization of the best method focused on finding the minimal time needed for fragmentation recovery and thus, to provide a useful protocol for optimized soft coral fragmentation.

# MATERIALS AND METHODS

## Sample species and reproduction methods

The soft coral used in the experiment, *Xenia umbellata* Lamarck, 1816, originated from the northern Red Sea and has been kept in recirculating holding tanks for approximately 3 years.

Four methods were tested in this experiment, *i.e.*, the glue method, rubber band method, tunnel mesh method, and plug mesh method as described below. Large colonies were carefully collected from the holding tank and fragmented into smaller sizes in a temporary water basin filled with water from the holding tank. To minimize the stress on fragments, no more than five fragments were treated at the same time in the water basin. Once five fragments were ready, they were attached on calcium carbonate coral plugs (Aquaforest® AF Plug Rocks cuttings) or placed in egg crates depending on the method and placed into designated tanks immediately. After the water in the basin was replaced, this procedure was repeated until the needed number of fragments was fulfilled. All fragmentation procedures and further assessments (see Reproduction method assessment categories) were conducted by a single individual.

### The glue method

One coral fragment was glued to the center of a dry coral plug using a few drops of coral glue (Microbe-Lift Coralscaper seconds glue). The lower part of the cut coral was embedded into the glue substrate (Fig. S1) using tweezers and left to harden for a few seconds in the water basin before being transferred to its respective tank.

### The rubber band method

Each coral fragment was attached to the coral plug using two rubber bands, with a diameter of 2 cm when unstretched, crossing each other making an 'X' shape on the plug (Fig. S2). Rubber bands were applied as loose and gently as possible to minimize the stress caused on fragments by letting minimum force to keep the fragments onto the plugs. Fragments were then transferred to respective tanks immediately after attachment.

### The tunnel mesh method

Adapted from the tunnel cube method by *Curry (2020)*, the tunnel mesh method was composed of two mesh fabrics and two egg crate grids held together with 2–4 rubber bands (Fig. 1A). The mesh fabrics had a diameter of 3.5 mm (Large mesh: Mesh L) and 2.0 mm

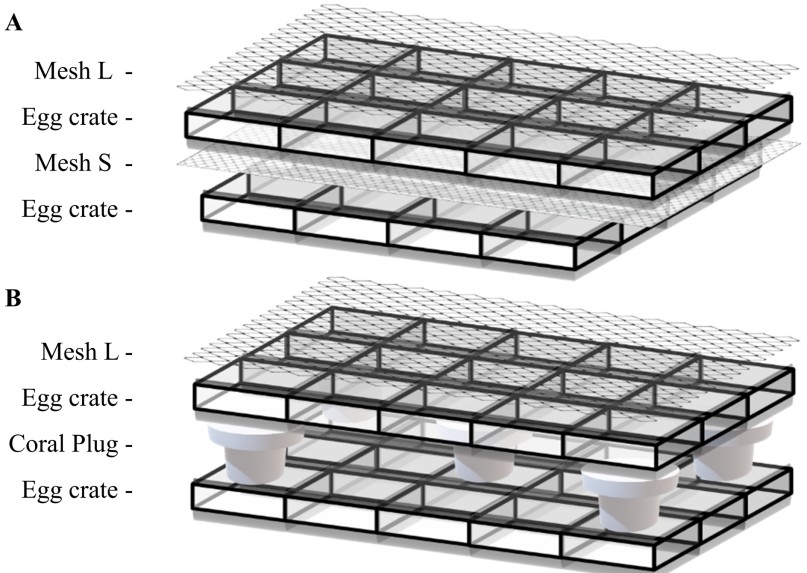

**Figure 1 Illustration of (A) tunnel mesh method and (B) plug mesh method.** All components are fixed with two to four rubber bands on the sides. An extra egg crate is an important factor so that there can be water circulating through the egg crate holes. Once the coral fragment is attached to the mesh fabric or plug beneath, the egg crate and mesh fabric on top can be removed. Mesh fabrics are differentiated by diameter size of 3.5 mm (Large mesh: Mesh L) and 2.0 mm (Small mesh: Mesh S) per opening.

(Small mesh: Mesh S) per opening. Different mesh fabrics can be used as long as they serve their purpose. The top mesh (Mesh L) needs to keep the coral fragments inside the egg crate tunnel while maximizing the water flow. The bottom mesh screen (Mesh S) needs to have a mesh size small enough for the fragments to attach while subsequently allowing water to flow through. A fragmented coral was placed into one egg crate tunnel and closed off with Mesh L (Fig. 1A). After assembling all the components with rubber bands, the tunnel mesh grid was put in its respective tank. Coral fragments were placed in egg crate tunnels without fragments in neighboring tunnels to prevent physiological interference to each other.

In the beginning of the experiment, to keep the tunnel mesh method assembly from floating, empty plugs were placed on top which later on served as control plugs to measure the blank control in oxygen flux assessments (see *Oxygen fluxes*).

### The plug mesh method

Also adapted from the tunnel cube method by *Curry (2020)*, the plug mesh method required two egg crate grids, one mesh fabric (Mesh L) and five coral plugs (Fig.1B). Five coral plugs were evenly placed in the lower egg crate to hold the plugs with another egg crate placed on top of the plugs to hold the coral fragments. One fragmented coral was placed inside the top egg crate tunnel covering the coral plug. Once all fragments were in place, a Mesh L fabric was placed to cover the structure, which was subsequently fixed with rubber bands. The whole assembly was then transferred to its respective tank. Fragment arrangement was identical to the tunnel mesh method described previously.
## Experimental design and maintenance

We used four tanks with a total water volume of approximately 50 l each. After filling the tanks with water from the holding tank, they were left to stabilize for 48 h. A total of 80 fragments were evenly divided over the four tanks. Thus, each tank contained a total of 20 newly fragmented corals (five fragments x four methods) with two additional control corals originating from the same holding tank as the newly fragmented corals.

Water parameters were then maintained at the following levels, similar to the holding tank, throughout the experiment: salinity 34–36 PSU, temperature 25–26.5 °C, pH 8.26 ± 0.02, calcium 415 ± 5 mg L$^{-1}$, magnesium 1,230 ± 19.64 mg L$^{-1}$, nitrite <0.01 mg L$^{-1}$, nitrate <0.5 mg L$^{-1}$, ammonium <0.05 mg L$^{-1}$, phosphate <0.02 mg L$^{-1}$. Water flow (300 L/h) was provided by EHEIM CompactOn 300 pumps (EHEIM GmbH and Co. KG, Deizisau, Germany). Light was provided by daytime LED light (WALTRON daytime® LED light, Germany) in a 9:15 h light:dark cycle with an intensity of ~55 ± 5 μmol m$^{-2}$ s$^{-1}$. Approximately 10% of water was exchanged each day with water from the holding tank. Coral fragments were fed with coral food (REEF-ROIDS$^{TM}$ Engineered Coral Food, Polyp Lab, Canada) 2 to 3 times per week.

For the optimization process, we used three tanks with a total of 15 newly fragmented corals. Each tank had five fragments and two controls from the same holding tank. All the other parameters were identical as the main experiment.

## Reproduction method assessment categories

We based our assessment of these four methods on (1) success rate, (2) fragmentation time, (3) oxygen fluxes, and (4) pulsating rate. Success rate, defined as the attachment percentage to the desired surface of each method, and fragmentation time were used to determine the time- and labor- efficiency of each method, while oxygen fluxes and pulsating rate measurements were used as a proxy for the health status (*McGillis et al., 2009*; *Jantzen et al., 2013*; *Vollstedt et al., 2020*; *Simancas-Giraldo et al., 2021*). The method with the best results was then further optimized using three assessment categories: (1) success rate, (2) oxygen fluxes, and (3) pulsating rate. Compared to the main experiment, assessments were conducted in a more detailed timeline to provide a better understanding and thus, to optimize the best method.

### Success rate

Success rate was assessed and calculated on the eighth day post fragmentation as there were no further changes observed nor made to the fragments after the eighth day, *e.g.*, cutting rubber bands or scraping of egg crates.

For the tunnel mesh and plug mesh method, a successful attachment was counted when it was attached to the mesh or plug, respectively, not the egg crate. Also, it was considered to be a successful attachment only if the coral fragment remained attached to the mesh or plug under ambient water flow without support from the rubber band or egg crate tunnel. If fragments attached to the egg crate walls, they were gently scraped

off and placed back on the mesh fabric or plug with the polyps facing upwards to stimulate attachment. Note: Failure rates do not represent mortality. Rather, they indicate that the attachment to the desired surface has failed.

### Fragmentation time

While fragmenting the corals, the time needed to produce one fragment assembly was recorded. This included the time for cutting, mounting, and placing one fragment assembly into its designated tank.

### Oxygen fluxes

Measurements were done on the twelfth-day post fragmentation using 19 jars (total = 19, 3 per method, 3 controls, 3 empty plugs as blank control, and 1 with only water to measure background oxygen evolution/depletion).

Coral fragments were incubated for 1 h in light to assess net photosynthesis ($P_{net}$) and 1 h in dark conditions to assess respiration ($R_{dark}$). Coral fragments were transferred into individual 160 ml glass jars filled with the same water from where the coral came from containing a small magnet for stirring. Right after one fragment was transferred into a jar without any air contact, the time, temperature, and oxygen concentration were measured using an optode sensor (Hach IntelliCAL/Optical Dissolved Oxygen Probe or similar) and then sealed air-tight making sure there were no air bubbles inside. Jars were placed in a water bath kept at 26 °C which was placed on top of a magnetic stirring machine (Poly 15, Thermo Scientific VARIOMAG® Magnetic Stirrers) set at 190 rpm to prevent a gradient of oxygen forming inside the jars. After the 1 h incubation, the lid was opened carefully and the temperature and oxygen level were measured again. The difference in oxygen concentration between start and finish was used to calculate the oxygen fluxes. Data was corrected for controls, and normalized to incubation time and number of polyps per sample resulting in fluxes presented in "μg $O_2$ polyp$^{-1}$ h$^{-1}$". Light incubations were performed under the same light conditions as in the respective tanks while dark incubations were performed in complete darkness. Gross photosynthesis ($P_{gross}$) was calculated using the following equation: $P_{net} = P_{gross} + R_{dark}$ (e.g., *Jantzen et al., 2013*; *Tilstra et al., 2019*). Oxygen flux measurements for optimizing the best method were measured on the seventh-day post fragmentation using 10 jars (total = 10, three controls, three respective method, three empty plugs as blank control, and one with only water to measure background).

### Pulsating rate

Pulsating rates were assessed by counting pulsation of three randomly selected polyps per fragment for 30 s. This was then calculated to give a mean value of the number of pulsations per minute (pulses min$^{-1}$). A full contraction and opening of all polyp tentacles were considered as one pulse. Pulsating rates were assessed every 48 h at a fixed time to avoid differences due to circadian rhythms.

In the optimization of the best method, measurements were done every 24 h for 7-days to provide more detailed information.

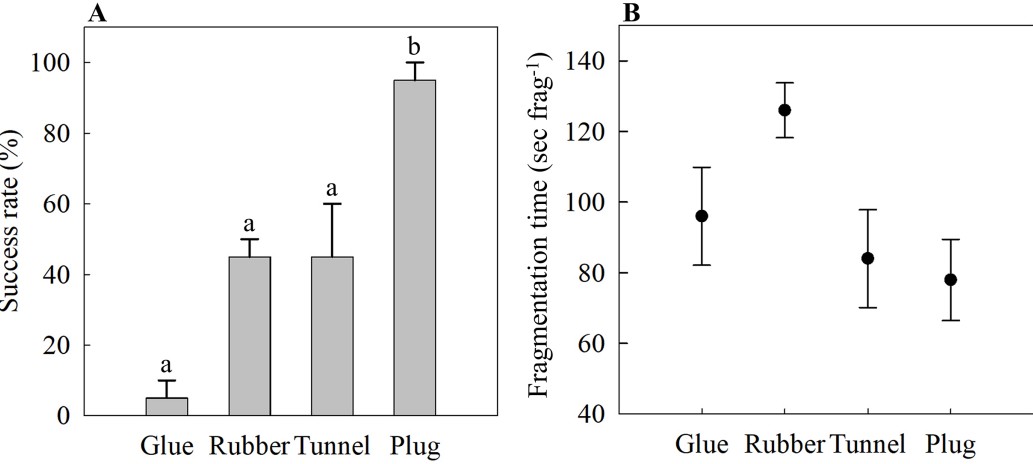

**Figure 2 (A) Attachment success rate and (B) fragmentation time of four different fragmentation methods as an assessment of time- and labor-efficiency.** Four methods are the glue (Glue), rubber band (Rubber), tunnel mesh (Tunnel), and plug mesh method (Plug). Values are given as means ± SE of $N = 3$–4 replicates. Different letters (a, b) indicate significant differences between the methods (one-way ANOVA, $F_{[3,12]} = 21.75$, $p < 0.001$).

## Statistical analyses

Statistical analyses were performed using Sigmaplot 14.5 and R Studio for Windows. All data are presented as means ± SE. Differences between treatments were analyzed for significance by one-way analysis of variance (ANOVA) using "method" as the main factor. Percentage data were transformed by applying the arcsine square root transformation for the one-way ANOVA test. Post-hoc analysis (Tukey's HSD) was carried out when significant differences were found. Asterisks indicate significant differences between methods (*$p < 0.01$, **$p < 0.001$).

## RESULTS

### Main experiment

#### *Success rate*

The glue method had the lowest success rate of 5 ± 5%, while the rubber band and tunnel mesh method had a success rate of 45 ± 5% and 45 ± 15%, respectively (Fig. 2A). The plug mesh method had a significantly higher success rate compared to the other three methods (95 ± 5%; one-way ANOVA, $F_{[3,12]} = 21.75$, $p < 0.001$; Fig. 2A).

#### *Fragmentation time*

The time needed to assemble one coral fragment was not statistically significant (one-way ANOVA, $F_{[3,12]} = 3.17$, $p = 0.064$). The glue method needed 96 ± 14 s frag$^{-1}$ while it took the longest for the rubber band method of 126 ± 8 s frag$^{-1}$ (Fig. 2B). The tunnel mesh method took 84 ± 14 s frag$^{-1}$ while the plug method required the shortest time, 78 ± 11 s frag$^{-1}$, to produce one fragment assembly (Fig. 2B).

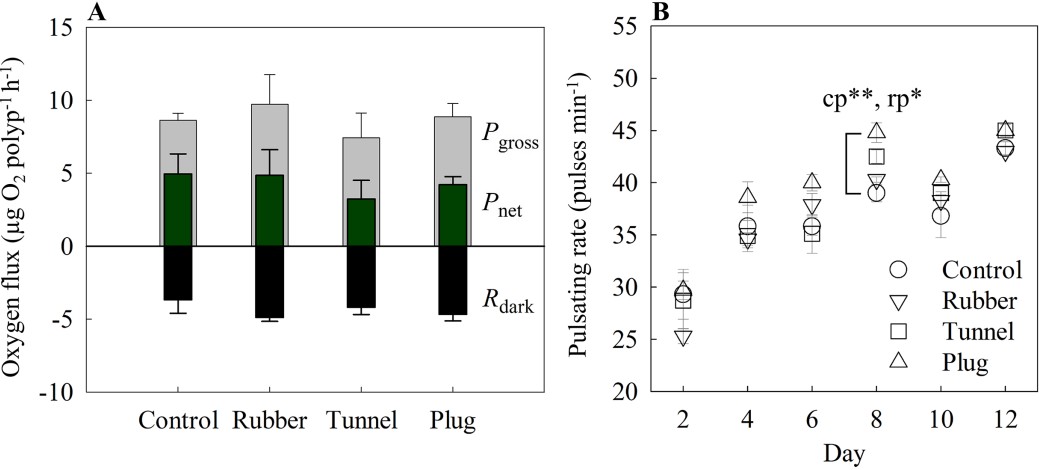

**Figure 3** (A) Oxygen fluxes and (B) pulsating rates measured on the twelfth-day post fragmentation as health assessments of three different fragmentation methods. The three methods are the rubber band (rubber), tunnel mesh (tunnel), and plug mesh method (plug). $P_{gross}$ = gross photosynthesis, $P_{net}$ = net photosynthesis, $R_{dark}$ = dark respiration. Values are given as means ± SE of $N$ = 3-4 replicates. Asterisks ($^*p < 0.01$, $^{**}p < 0.001$) indicate significant differences between the methods, while letters (c: Control, p: plug mesh method, r: rubber band method) indicate between which methods they differed in. Note: The glue method was not assessed as only one out of 20 replicates successfully attached to the plug.

### Oxygen fluxes

There were no statistically significant differences between the control and each method for all three parameters measured, *i.e.*, $P_{gross}$ (one-way ANOVA, $F_{[3,8]} = 0.45$, $p = 0.725$), $P_{net}$ (one-way ANOVA, $F_{[3,8]} = 0.36$, $p = 0.786$), and $R_{dark}$ (one-way ANOVA, $F_{[3,8]} = 0.84$, $p = 0.509$) (Fig. 3A). Mean values ranged between 7.43 to 9.73 µg $O_2$ polyp$^{-1}$ h$^{-1}$ for $P_{gross}$, 3.25 to 4.95 µg $O_2$ polyp$^{-1}$ h$^{-1}$ for $P_{net}$, and −4.88 to −3.68 µg $O_2$ polyp$^{-1}$ h$^{-1}$ for $R_{dark}$ (Fig. 3A).

The glue method was not assessed here as only one out of 20 replicates successfully attached to the plug.

### Pulsating rate

With the exception of day 8 (one-way ANOVA, $F_{[4,12]} = 9.76$, $p < 0001$), overall pulsating rates differed in every measurement day without significant differences among methods. Among the methods, the plug mesh method had significantly higher pulsating rates than the control and rubber band method (Tukey's HSD, $p < 0.001$, $p < 0.01$, respectively). All methods had lower average pulsating rates in the beginning of the experiment which gradually increased. On the second day, the mean pulsating rate of all methods was 27.13 ± 1.01 pulses min$^{-1}$ while from day 4 to 12, the mean pulsating rate of the three methods was in the range of 35.99 to 44.06 pulses min$^{-1}$ (Fig. 3B).

The glue method was not assessed here as only one out of 20 replicates successfully attached to the plug.

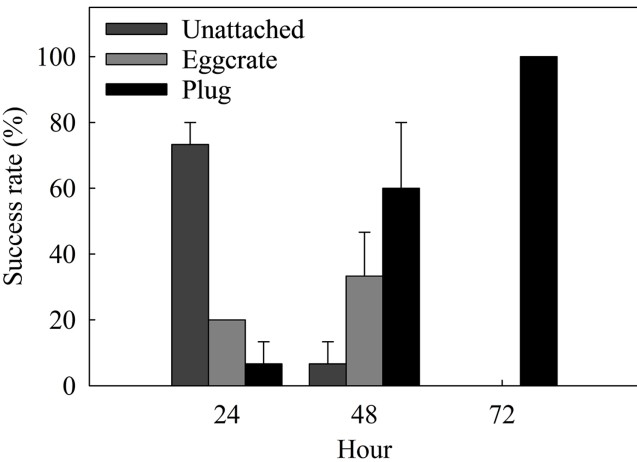

**Figure 4 Attachment success rates of the best assessed method, the plug mesh method, over a period of 72 h.** Fragments are either unattached to anything (Unattached: dark gray) or attached to the egg crate wall (Egg crate: light gray) or the plug (Plug: black). Values are given as means ± SE of $N = 3$ replicates.

## Optimization of best method

### Success rate

The success rate of the best method, the plug mesh method, was assessed in a more detailed time scale. Coral fragments started to attach to either the egg crate (20%) or the plug (6.67 ± 6.67%) after 24 h of fragmentation (Fig. 4). From the ones unattached to anything (73.34 ± 6.67%), a thin layer of dead tissue could be observed falling off of the cut surface. More than half of the fragments were able to attach to a surface by 48 h and after 72 h, 100% of the fragments were attached to the plug.

### Oxygen fluxes

Both $P_{gross}$ and $R_{dark}$ differed significantly between the control and the plug method (one-way ANOVA, $F_{[1,4]}$ = 22.52, 31.79, $p$ = 0.009, 0.004, respectively). There were no significant differences for $P_{net}$ (control = 3.78 ± 0.57 μg $O_2$ polyp$^{-1}$ h$^{-1}$, plug = 6.19 ± 0.86 μg $O_2$ polyp$^{-1}$ h$^{-1}$; one-way ANOVA, $F_{[1,4]}$ = 5.48, $p$ = 0.079) (Fig. 5A).

### Pulsating rate

On the first day, the pulsating rates were significantly lower in the fragmented plugs compared to the control (one-way ANOVA, $F_{[1,13]}$ = 22.84, $p < 0.001$). After 2 days, the newly fragmented coral frags had no significant difference in mean pulsating rate as the control (control = 41.78 ± 0.59 pulses min$^{-1}$, plug = 37.24 ± 0.92 pulses min$^{-1}$; one-way ANOVA, $F_{[1,15]}$ = 4.89, $p$ = 0.043) and also for day 3 and 4 (one-way ANOVA, $F_{[1,16]}$ = 0.3, 0.43, $p$ = 0.54, 0.524, respectively) (Fig. 5B). After 5 days of fragmentation, the mean pulsating rate was significantly higher for the fragmented coral frags (control = 43.22 ± 0.29 pulses min$^{-1}$, plug = 45.20 ± 0.29 pulses min$^{-1}$; one-way ANOVA, $F_{[1,16]}$ = 8.90, $p$ = 0.009) and remained significantly higher on both day 6 and 7 (one-way ANOVA, $F_{[1,16]}$ = 23.90, 22.96, $p < 0.001$, $< 0.001$, respectively).

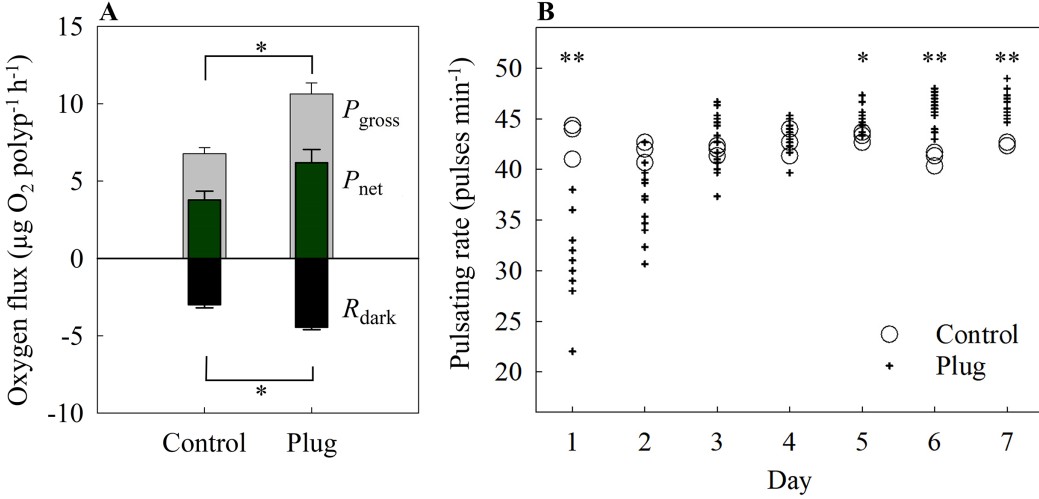

**Figure 5** **(A) Oxygen fluxes and (B) pulsating rates of the plug mesh method.** Oxygen fluxes were measured after 7 days of fragmentation while pulsating rates were measured every 24 h starting from 24 h after fragmentation as health assessments. $P_{gross}$ = gross photosynthesis, $P_{net}$ = net photosynthesis, $R_{dark}$ = dark respiration. Values of bars are given as means ± SE of $N$ = 3 replicates. Asterisks (*$p < 0.01$, **$p < 0.001$) indicate significant differences between treatments. Top bracket applies to $P_{gross}$ only.

## DISCUSSION

While fragmentation methods for hard corals are abundant and well described, there is a lack of fragmentation techniques for studying the second biggest benthic group found on coral reefs, namely soft corals. Here, we present an assessment and comparison of four different methods for soft coral reproduction. Our results show that of the four assessed methods, the plug mesh method, which is adapted from the tunnel cube method (*Curry, 2020*), is the most time- and labor-effective method for easy reproduction of the soft coral *X. umbellata*.

Oxygen fluxes and pulsation rates were measured as a proxy for health of the *X. umbellata* coral fragments (*McGillis et al., 2009*; *Jantzen et al., 2013*; *Vollstedt et al., 2020*). However, no significant differences were found between the methods and the control corals, suggesting that all fragments were at their respective optimal health. As such, "success rate" and "fragmentation rate" were the main parameters used for the assessment of the "best method".

Out of the four assessed methods, the glue method, which works well for hard corals (see *e.g.*, *Tilstra et al., 2017*), demonstrated the lowest success rate. Only 1 out of 20 fragmented colonies attached to the coral plug, albeit on the side of the plug. This failure to adhere to the glue could be explained by the formation of a dead tissue layer observed after fragmentation or coral derived mucus observed during fragmentation between the glue and the healthy coral tissue.

Similar to the glue method, the rubber band method damaged the coral fragments after fixation on the plug despite lowering the pressure of the rubber bands on the coral tissue to a minimum. Still, various fragments were sliced in half due to the pressure put on them by the rubber bands. The fragments marked as 'fail' due to slicing or no

attachment to the plug surface managed to attach to either the rubber band or onto different surfaces similar as the failure fragments of the glue method, also resulting in no mortality. However, the success rate of the rubber band method still exceeded the glue method and was similar to the success rate of the tunnel mesh method (Fig. 2A). Although the rubber band method appeared to be less time efficient (though not significant, Fig. 2B), this method is preferred over the tunnel mesh method as the mesh cannot be separated from the coral fragment when fixed to a plug. This is important because algae tend to grow with ease on materials such as cotton fabrics, which can interfere with experiment results (*Christenson & Sims, 2012*; *Gross et al., 2013*). Furthermore, while the coral fragments will indeed first attach to the mesh fabric, eventually they will attach to a more stable surface, such as the tunnel cubes (*Curry, 2020*) or the egg crate used in the present study. This may also explain the high success rate for the plug mesh method (Fig. 2A). Despite initial attachment to the egg crate for some of the coral fragments, scraping them off the egg crate wall forced the fragments to finally settle onto the plugs. This resulted in the highest success rate among the four assessed methods (Fig. 2A). Additionally, when considering both success rate and fragmentation time, the plug mesh method was the most labor- and time-efficient method for fragging the soft coral *X. umbellata*, even after considering the additional time invested for each method before and after fragmentation, *e.g.*, mesh fabric preparation, reattachment processes, and scraping off egg crate walls. Finally, considering faster recovery in pulsating rates, though non-significant (Fig. 3B), and the reusability of all materials we conclude that the plug mesh method is the preferred fragmentation technique for *X. umbellata*.

In the optimization process of the plug mesh method, more detailed data was acquired to assess the attachment and recovery speed of fragments. Attachment success rates suggest that fragments can attach to a surface within 24 h after fragmentation. However, all fragments were firmly attached to the plug 72 h post fragmentation and was thus considered the optimum time needed (Fig. 4). Attachment increased considerably after a thin layer of dead tissue was observed and removed 24 h post fragmentation. This dead tissue layer is likely due to epithelialization, a process where epithelial cells move upwards and repair the wounded area (*Meszaros & Bigger, 1999*). Moreover, a study that assessed the attachment abilities of the soft coral *Dendronephthya hemprichi* reported that root-like processes (RLPs), which play a major role in the attachment process, developed after 4 days of fragmentation (*Barneah, Malik & Benayahu, 2002*). We speculate that the xeniids, on the other hand, were able to attach much faster than the RLPs developing *D. hemprichi* as they utilize a natural, yet unknown, adhesive material during or after epithelialization has taken place (*Ellis & Sharron, 1999*; *Callow & Callow, 2002*; *Bromley & Heinberg, 2006*). Further investigation in comparing different healing rates and attachment methods of an array of soft corals could shed more light on this.

Pulsating rates showed that a minimum of 48 h was needed for fragments to recover, while after 6 days of fragmentation, pulsating rates were significantly higher than the control (Fig. 5B). A similar pattern was observed during the main method assessment

(Fig. 3B). Correspondingly, the $P_{gross}$ and $R_{dark}$ were also significantly higher for the fragments than the control (Fig. 5A). Whether this is the result of size difference, subsequent self-shading, or a specific defense mechanism of the fragmented corals remains to be determined. Some studies have demonstrated positive effects of pulsation of the *X. umbellata* as this increased gas exchange between the coral and the surrounding water, thereby increasing energy and nutrient acquisition (*Kremien et al., 2013*; *Wild & Naumann, 2013*). As such, increased pulsation rates could facilitate recovery of the wounded fragment by assimilating more resources. Indeed, higher pulsation rates could increase $P_{net}$ and thus fixation of C as observed in the xeniid coral *Heteroxenia fuscescens* (*Kremien et al., 2013*). However, no significant differences were found in $P_{net}$ in the current study. Nonetheless, higher $R_{dark}$ and $P_{gross}$ associated with fragmented corals compared to controls suggests an enhanced metabolism, which supports the hypothesis that fragments were producing and utilizing more resources, possibly for wound healing. Whether this enhanced metabolism is solely due to increased autotrophy or, in combination with, increased heterotrophy remains to be determined.

## CONCLUSION

In conclusion, our results show that the plug mesh method is an easy, highly time- and labor-efficient fragmentation method for the *X. umbellata* soft coral that could potentially be used as a base for conducting essential research on diverse soft corals. It has significance in that the plug mesh method makes propagating soft corals rather effortless, thus, leading to higher probabilities of *ex situ* reproductions rather than harvesting from the wild. This could potentially result in reduced disturbances of coral reef communities. The plug mesh method will also benefit laboratories using soft corals as study specimens by greatly reducing the preparation time needed to produce coral fragments, possibly resulting in more studies based on soft corals. As the second most abundant benthic group of organisms on coral reefs, research on soft corals will increase our understanding of their role in these complex ecosystems which may lead to a better understanding and/or implementation of reef rehabilitation, conservation, and managing efforts.

## ACKNOWLEDGEMENTS

We thank Selma Mezger, Julia Plewka, and Lisa Zimmermann for help with setting up the experiment. A step-by-step protocol of the plug mesh method has been uploaded to protocols.io (*Kim, Wild & Tilstra, 2021*).

### Funding

Financial support was provided by Bremen university baseline funds to Christian Wild and the German Research Foundation (DFG) grant no. Wi 2677/16-1 to Christian Wild. The funders had no role in study design, data collection and analysis, decision to publish, or preparation of the manuscript.

## Grant Disclosures

The following grant information was disclosed by the authors:
Bremen university.
German Research Foundation (DFG): Wi 2677/16-1.

## Competing Interests

The authors declare that they have no competing interests.

## Author Contributions

- Sohyoung Kim conceived and designed the experiments, performed the experiments, analyzed the data, prepared figures and/or tables, authored or reviewed drafts of the paper, and approved the final draft.
- Christian Wild conceived and designed the experiments, authored or reviewed drafts of the paper, and approved the final draft.
- Arjen Tilstra conceived and designed the experiments, authored or reviewed drafts of the paper, and approved the final draft.

## Patent Disclosures

The following patent dependencies were disclosed by the authors:
Noel Thomas Curry
US 20200128797A1 16/174,280
Apr. 30, 2020
United States Patent Application Publication

## Data Availability

The raw measurements are available in the Supplemental File.

## Supplemental Information

Supplemental information for this article can be found online at http://dx.doi.org/10.7717/peerj.12589#supplemental-information.

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
