# Peer review of "Effective asexual reproduction of a widespread soft coral: comparative assessment of four different fragmentation methods"

_PeerJ, doi:10.7717/peerj.12589_

## Round 0.1 · original submission · Major Revisions

Based on the feedback of the referees, I am asking you to undertake a major revision to your submission. As you can see below, we now have comments from two expert referees who have offered a variety of suggestions for improvement of your manuscript. The first asks for better rationale and justification for the value of propagating xenids, and the second questions whether the results seen in aquaria are relevant to a field setting. I would add that I discussed your submission with the editorial staff for the the special issue https://peerj.com/special-issues/85-coral-reef-restoration, and while all agreed that your manuscript could be refocused to become an appropriate fit to the issue, it was not written with this special issue in mind for the initial submission and could use some revision to improve the relevance and discussion of the topic of active reef conservation and restoration. Thus, if you would like the article to be considered for that issue, please note that in your rebuttal letter, and make an effort to include the scope from the call more explicitly in your revised manuscript. So, although I do not expect that the referee concerns will be difficult for you to address, they do involve some substantial editorial changes to the manuscript and together with a revision to better fit the call for papers tips this toward a more major revision. If you do not wish to submit to the special issue, that is also acceptable. Please note that decision in your response letter, and we will consider your paper as a typical submission to the journal instead.

Finally, I would note here that it is PeerJ policy that additional references suggested during the peer-review process need only be included if the authors are in agreement that they are both relevant and useful.

If you decide to undertake the suggested revisions, please ensure that all reviewer comments are addressed in a rebuttal letter that outlines how you have addressed each comment. Any edits or clarifications mentioned in the rebuttal letter should also be inserted into the revised manuscript where appropriate. It is a common mistake to address reviewer questions in the rebuttal letter but not in the revised manuscript. If a reviewer raised a question, then your readers will probably have the same question so you should ensure that the manuscript can stand alone without the rebuttal letter. Directions on how to prepare a rebuttal letter can be found at: https://peerj.com/benefits/academic-rebuttal-letters/ if you need additional guidance.

Please do not hesitate to contact me if you have any questions, and I look forward to seeing your revised manuscript.

Reviewer 1 ·

Basic reporting

See ahead

Experimental design

See ahead

Validity of the findings

See ahead

Additional comments

I have carefully read the manuscript by Kim et. al. entitled: “Effective asexual reproduction of a cosmopolitan soft coral: Comparative assessment of four different fragmentation methods”.
The manuscript presents experimental approaches for asexual propagation of a soft coral. I praise the authors for putting together the results on this subject. However, I have some critical comments addressed below:
1. I wonder what is the scientific evidence for claiming that X. umbellata is a cosmopolitan
soft coral. Google says for this term: “A plant or animal found all over the world”. This is definitely (!) not the case for this soft coral. The mistake should be corrected and the current biogeographic distributional boundaries of the study species has to be addressed correctly with appropriate references.
2. The rational of the study should be addressed from a different perspective which unfortunately has been ignored in the manuscript. Xeniid soft corals have been found to be invasive, replacing other benthic reef organisms including stony corals included, following reef deterioration etc. The recent literature indicates that xeniids have become dominant benthos in places where that did not exist before. In addition, studies have shown that they tolerate global change conditions and shift in community structure is expected on may reefs to become soft coral dominated. All these topics have been addressed in the scientific literature, but unfortunately not mentioned in the manuscript.
3. The above (2.) may challenge the concept presented in the manuscript arguing the environmental and conservation arguments ask for adequate asexual propagation methods of xeniids. But, why to transplant them on denuded reefs if they are considered to be successful colonizers as well as fugitive species? In addition, their prolonged reproductive season and high fecundity as shown in several studies, but not mentioned in the current ms, significantly contribute to the immense success and even massive space occupation on the reef.
4. The hobby reef aquarists usually want to control Xenia spread in their tanks. Their successful development in the tanks also diminish the practical value of the current results. I wonder why the authors did not refer to this point in the text.
5. Unlike nephtheid soft corals, xeniids have no RLPs. Their attachment to hard substrate is conducted by natural glue, yet of unknown nature
6. The authors missed several studies on soft coral husbandry which are so relevant to their study (e.g. Aquaculture Research, 2010, 41, 1748^1758, Natural Resources Vol.07 No.06(2016), many others on the red coral and other gorgonians). A thorough review of such references is required in order to frame the current study and to pin point its novelty.

Reviewer 2 ·

Basic reporting

The language is clear and relevant literature is referenced.

Experimental design

The experimental design is clearly described and the research question and the purpose of the study are well defined. Methods are effectively described and could be replicated.

Validity of the findings

The author cited relevant research that shaped the design of the experiment and in support of findings. It remains unclear how effective this practice will be in a field setting, but the purpose of the study is clear and the targeted questions are effectively tested and answered through the lab trials.

Additional comments

While the findings of this lab-based study suggests that the plug mesh method is the most effective approach, it should be considered that this may only be true in an aquarium setting. Without having run field trials it may be difficult to determine the challenges and success involved with using this approach on a coral reef, where animals, surge, and other environmental disturbances could influence the effectiveness of this method. It should also be considered that best practices for restoration involve minimizing the amount of artificial materials left on a reef, so its possible that one of the methods that appeared to be less successful in the lab setting may be more suitable for the field setting.

It is unclear if the measurement for fragmentation time was completed by a single individual or multiple participants who could have varying levels of experience in this type of skill.

If oxygen fluxes and pulsation rates served as a proxy for health, then survivorship should be one of the parameters being evaluated in this study. Although it was mentioned that mortality occurred among some of the fragments, it is a significant indicator in measuring the success of fragmentation and attachment method, especially considering that mortality was so high among the glue method group.

---

## Round 0.2 · accepted · Accept

Thank you for your revisions and careful consideration of the referee comments. Given that you are not aiming for the special issue on active reef restoration, it seems to me that the revisions needed to satisfy the referees was relatively minor and the major revision suggested to fit the scope of the special issue is not needed. Reading through your revised manuscript and the fact that your incorporated all the suggested revisions made by the expert referees, I see no reason to waste anyone's time with requesting another round of review. I am happy to accept your revised manuscript, congratulations.